# Design and Implementation of Telemedicine System Using Light Fidelity and PIC16F877A Microcontroller

M.R.Ezilarasan

*Department of Electronics and Communication Engineering*
*Vel Tech Rangarajan Dr. Sagunthala R&D Institute of Science and Technology*
Chennai, India
drezilarasan@veltech.edu.in

Man-Fai Leung

*School of computing and Information Science,Faculty of science and Engineering*
Anglia Ruskin University, Cambridge, United kingdom
Man-fai.leung@aru.ac.uk

Xiangguang Dai*
*Chongqing Engineering Research Center of Internet of Things and Intelligent Control Technology,*
Chongqing Three Gorges University, Chongqing, China,
daiziangguang@163.com

*Abstract—* **Medical body area networks currently use wireless communication technology to give patients and carers more freedom and convenience and radio frequency (RF) is the existing medium in healthcare applications. In this the possibility of electromagnetic waves interfering with precision medical equipment still exists, though. This study uses the newly developed wireless visible light communication (VLC) technology to provide a novel design and implementation of a medical healthcare information system. VLC is also referred as light fidelity (Li-Fi). Visible light-emitting diodes (LEDs), which are expected to overtake traditional incandescent and fluorescent lights as the dominant lighting source in the near future owing to their energy efficiency, can be used with VLC. In this research the patient's health is monitored using sensors, and an LCD will display the results. An LED lightbulb will communicate the data in the form of light. The photo-detector at the receiving end gathers the transmitted data, shows the output on an LCD, and informs the carers by beeping every 15 seconds for respiration and 30 seconds for heartbeat. These lighting fixtures can function as wireless data transmission devices in addition to sources of illumination by utilising the fast switching power of LEDs. Hospital regions with restricted radio frequencies can benefit from data services and monitoring provided by the prototype VLC-based medical healthcare system.**

Keywords— VLC, Li-Fi, Healthcare, Optical wireless communication, LED, Photodetector.

## I. INTRODUCTION

Over the last ten years, optical wireless communication (OWC) has drawn a great deal of attention. OWC is viewed as an advantageous and complementary communication method to traditional radio frequency (RF) technology [1], which operates within a licensed and regulated electromagnetic spectrum band between 30 kHz and 300 GHz. Electromagnetic spectrum range is shown in below figure 1. Electromagnetic spectrum has different frequency ranges and for applications.

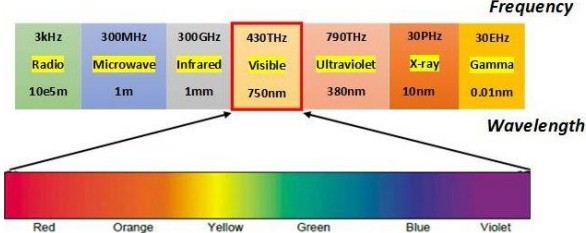

Figure 1 VLC frequency spectrum

This spectrum, commonly used for various wireless communication services such as mobile phones, Wi-Fi, and Bluetooth, based on their frequency ranges and it is experiencing increasing challenges due to the exponential growth in wireless data traffic and the proliferation of sophisticated applications. The surge in demand for wireless data has led to spectral congestion, where the available bandwidth is insufficient to support the volume of data being transmitted. This congestion results in slower data rates, increased latency, and degraded overall performance of wireless networks. Traditional RF technologies, while effective, are struggling to keep up with the demand, leading to a so-called "spectrum crisis." This crisis is characterized by the saturation of the RF spectrum, making it difficult to achieve high maximum data rates and maintain reliable communication.

In contrast, OWC utilizes the optical spectrum, which is much broader than the RF spectrum and less congested. This allows OWC to offer higher data rates, lower latency, and improved performance in environments where RF communication may be less effective. For instance, OWC can be particularly advantageous in scenarios where RF interference is a problem or where high-density data transmission is required, such as in urban areas, office buildings, and industrial settings. Moreover, OWC can operate in unlicensed bands of the optical spectrum, providing greater flexibility and cost-effectiveness for deployment. It can be used in various applications, including indoor communication using visible light (often referred to as Li-Fi), outdoor communication for point-to-point links, and underwater communication where RF signals are highly attenuated.

Today, there is significance focus on researching new wireless communication alternatives that can provide massive connectivity, diverse data rates, low latency, high capacity, efficiency, and enhanced security [2]. In this context, VLC has emerged as promising wireless communication technique that could address key challenges within the wireless communications infrastructure [3]. VLC uses light that is visible as the transmission medium, leveraging LED's to transmit data. This technique offers several advantages over traditional RF-based systems. For instance, VLC has many advantages like high spped transmission, high data rate, security [4].

One of the most notable advantages of VLC is its potential to provide an alternative to the heavily congested RF spectrum, offering up to 10,000 times more capacity. Additionally, the VLC spectrum is unregulated and unlicensed, presenting a vast and untapped resource for data transmission. This makes VLC is a user friendly bandwidth solution that can help with the problem RF spectrum shortage. Moreover, VLC can be used in conjunction with other essential communication systems and devices without causing interference with the electromagnetic fields generated by RF devices. This characteristic is particularly valuable in sensitive environments such as airplanes and hospitals, where RF interference can pose significant risks. There are ongoing studies exploring the transmission of medical data, such as photo plethysmography, electrocardiography, and body temperature using VLC applications [5, 6].

Another key advantage of VLC is its inherent security featureDespite RF signals, visible light is suitable for highly secure connections where wireless data transfer is meant to stay within range of an access point because it cannot pass through walls and does not expand uncontrollably. This characteristic makes VLC an ideal solution for environments where data security is paramount. For example, in corporate offices and government facilities, where sensitive information is frequently transmitted, VLC can ensure that data remains confined within the physical boundaries of a room or building. This physical limitation drastically reduces the risk of eavesdropping and unauthorized access compared to traditional RF-based systems, where signals can be intercepted from a distance. In residential settings, VLC can offer secure communication for smart home devices, preventing potential hackers from accessing personal data transmitted over wireless networks.

Furthermore, VLC can be seamlessly integrated into existing lighting infrastructure, offering dual functionality in the same hardware. This dual-purpose capability means that buildings can have efficient lighting and secure data transmission without the need for additional installations, reducing costs and complexity. For instance, LED lights in a smart home could provide both illumination and secure network connectivity to various IoT devices, such as security cameras, smart locks, and home automation systems.

For all these reasons, VLC can be applied to most IoT-based smart systems [7]. In the context of smart cities, VLC can be used for vehicle-to-vehicle (V2V) and vehicle-to-infrastructure (V2I) communication, enhancing traffic management and safety. Intelligent transportation systems can leverage VLC to provide real-time data exchange between traffic lights and autonomous vehicles, reducing the risk of accidents and improving traffic flow. Additionally, VLC can be employed in smart grids for secure and efficient communication between various components of the electrical grid, facilitating better energy management and reducing the risk of cyberattacks on critical infrastructure. This research paper is organized as follows: Section 2 reviews the existing literature and previous studies relevant to our research, providing a comprehensive background and context while highlighting gaps and opportunities that our study aims to address. Section 3 delves into the concept of VLC, explaining its fundamental principles, advantages, applications, current state of development, and technical challenges. Section 4 presents our proposed methodology and approach, detailing the experimental setup, data collection methods, and analysis procedures, and outlining the innovative aspects of our work. Finally, Section 5 summarizes the key findings, discusses the implications and contributions to the field, and suggests potential areas.

## II. RELATED WORKS

Few more research in recent years, a revolutionary approach to wireless communication has emerged, known as the Internet of LED. This paradigm shift integrates the Internet of Things (IoT) with VLC using LED technology, opening up a realm of possibilities across various industries and applications. One of the most intriguing applications of this technology is seen in indoor navigation and art gallery monitoring is given in [8].In this a prestigious museum adorned with an array of LED lights embedded in the ceiling. These LEDs serve a dual purpose providing ambient illumination and acting as data transmitters. Through sophisticated algorithms and protocols, these LED arrays communicate with users' mobile devices, offering precise positioning information about products, exhibits, or artworks. This not only enhances the overall experience for visitors but also streamlines operations and enhances security in such environments.

The automotive sector has also embraced the Internet of LED with enthusiasm. Modern vehicles are equipped with advanced LED headlights and taillights, which are not just efficient in lighting up the road but also serve as integral components in vehicular VLC systems [9]. These systems utilize the rapid flickering capabilities of LEDs to transmit data, enabling real-time communication for collision prevention systems and enhancing overall road safety. While traditional wireless technologies heavily rely on RF systems, such as Bluetooth, Zig bee, WLAN, and WPAN, there's a growing realization of the limitations and potential risks associated with RF in certain environments, especially healthcare settings.

Concerns about electromagnetic interference and its impact on sensitive medical equipment and patient safety have spurred interest in VLC using energy-efficient LEDs as a viable alternative [10, 11]. LEDs offer a multitude of advantages over conventional light sources. Their long operating lifetimes, minimal power consumption, and exceptional reliability make them ideal candidates for applications requiring continuous and efficient data transmission [12].

Researchers are actively exploring how VLC using LEDs can not only address concerns about RF-related

interference but also revolutionize wireless communication in critical environments like hospitals and clinics. The ongoing research and development in the Internet of LED are poised to reshape the wireless communication landscape. From enhancing indoor navigation experiences to improving road safety and revolutionizing healthcare communication, the fusion of IoT with VLC using LED technology promises efficient, secure, and reliable connectivity across diverse sectors. As this technology continues to evolve, we can expect even more innovative applications and transformative impacts on how we communicate and interact in the digital age.

## III. VISIBLE LIGHT COMMUNICATION

Based on a numerical estimate of optical transmission needs, [13] University in Japan was the first to propose VLC, which uses white LEDs to transport data. Evolution of light is shown in figure 2

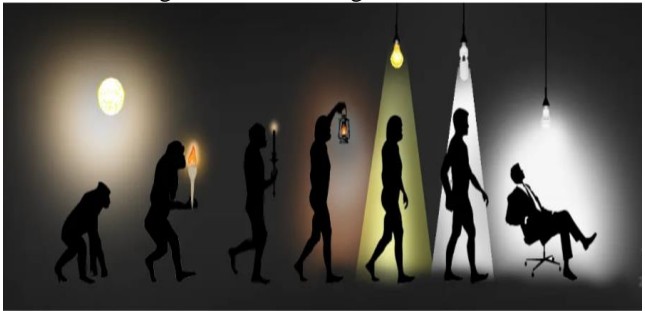

Figure 2 Evolution of Light [14]

Since then, a great deal of research has been done on the use of commercial white LEDs for short-range indoor communications. Owing to the characteristic of short-range communication restricted by the light beam's range, [15] suggested that VLC is more suited for location-based service applications, including giving consumers access to location-specific information. Additionally, new developments for VLC applications are always being made.

In [16] suggested a system of information that would allow medical facilities to use the hospital illumination network to offer private or public data services. Nevertheless, the study's primary focus was on the information system's architectural concepts; the actual design and implementation of medical data transfer via VLC has not yet been documented. In this work, we provide a safe and readily available substitute for RF wireless technology: a wireless VLC-based medical healthcare information system. VLC is a great option for wireless access in the medical and healthcare sectors where electromagnetic interference (EMI) and RF pollution are major concerns.

The greatest frequency utilized in RF technology is 10,000 times lower than the wide bandwidth of VLC [17]. Unlike the RF band, which is crowded and has issues with frequency allocation, the VLC band is currently unlicensed. Since communication takes place inside the visible light spectrum, the Federal Communications Commission (FCC) has not established any regulations. Due to the pre-installation of LED devices indoors and their compatibility with inexpensive electronic drive circuits, VLC-based systems also offer cost-effective installation.

Numerous problems with wireless RF networking methods were examined in [18]. Studies were conducted with an eye towards resolving these problems with VLC systems. In addition, a talk about applications, ways to solve current VLC problems, and upcoming advancements was given. In [19] Author has designed and implemented heterogeneous systems via wireless RF and VLC. A hybrid WiFi-VLC network makes up one system, while the other is created by aggregating Wifi and VLC in parallel utilising the Linux operating system's bonding approach. The downlink for the hybrid network is a VLC channel, which is only intended to be utilised in one way.

### A. VISIBLE LIGHT COMMUNICATION IN TELEMEDICINE.

Telemedicine refers to the use of telecommunications to transfer medical information and provide healthcare to patients remotely. The goal is to provide evidence-based medical treatment to anybody, anywhere, at any time. Research investigations and real-world deployments are increasingly using wireless technologies for telemedicine. It enhances the quality of care by facilitating the flexible collection of medical records and by giving users convenience and mobility. Numerous primary uses, including vital sign monitoring, electronic medical record maintenance, orders to carers, and non-medical services like entertainment, can be realised with the use of wireless technologies.

3.1 Proposed system design

Fig. 3 shows the entire system architecture for transferring multiple medical data. The transmitter module, the receiver module, the processing module, and the monitoring system are the four components that make up this system.

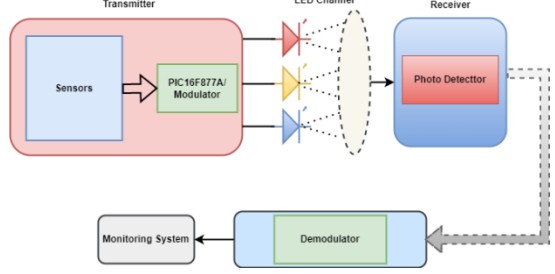

Figure 3. Proposed block diagram.

First, patients' biomedical signals—such as their temperature and heartbeat—are converted and sent via LEDs. In this case, it is assumed that every transmitter and receiver are perfectly synchronised. At the optical receiver, the data from the optical channel are transformed into a voltage signal after passing via a photodiode. The voltage signal was then detected in the monitoring system after demodulating to a digital signal in the processing module. Each step's specifics are listed below.

### B. Transmitter Section:

The transmitter section consists of several key components designed to ensure proper voltage regulation and sensor interfacing. Initially, a step-down transformer is used to convert the standard 230V AC supply to a lower voltage of 5V DC. This step-down transformer is crucial for safely powering the subsequent components. Following the transformer, a bridge rectifier is employed. The bridge rectifier's role is to convert the alternating current (AC) from

the transformer into direct current (DC), which is necessary for the operation of electronic components. To further refine the DC output and ensure a stable voltage, a voltage regulator, specifically the LM7805, is used.

The LM7805 maintains a consistent output of 5V DC, protecting the circuitry from voltage fluctuations that could potentially cause damage or unreliable operation. Additionally, a filter capacitor with a capacitance of 1000 microfarads (μF) is incorporated. This capacitor smooths out any remaining ripples in the DC output, providing a clean and steady voltage for the transmitter section. Various sensors, including heartbeat, temperature, and sound sensors, are connected to the PIC16F877A microcontroller. The PIC16F877A is a low-power, high-performance microcontroller that features 8KB of in-system programmable memory, allowing for flexibility in programming and updating. One of the standout features of the PIC16F877A microcontroller is its inbuilt Universal Asynchronous Transmitter/Receiver (UART). This UART module facilitates serial communication, enabling efficient data transmission between the microcontroller and other components or systems. Figure 4 illustrates the proposed block diagram for the transmitter section of the Healthcare Monitoring System (HMS) utilizing VLC This diagram provides a visual representation of how the components are interconnected and function together to monitor various health parameters.

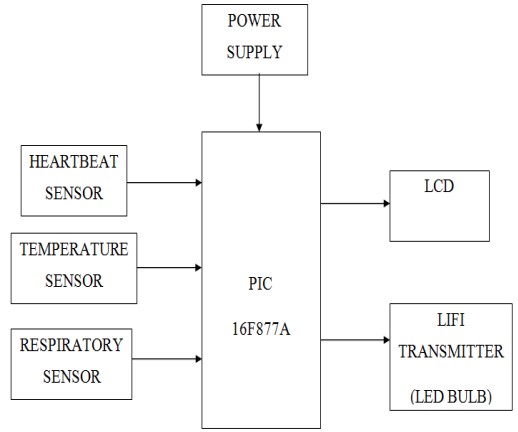

Figure 4. Block diagram of transmitter

Each sensor in the system generates its own analog or digital values, depending on the type of sensor. These values are then fed into the PIC16F877A microcontroller. This microcontroller plays a crucial role in analyzing the data received from the different sensors, processing it accordingly, and displaying the processed information on an LCD screen for easy monitoring and interpretation. In addition to handling sensor data, the PIC16F877A microcontroller is interfaced with a Li-Fi transmitter, which in this case is an LED bulb. Once the data from the sensors is processed by the microcontroller, the PIC establishes serial communication with the Li-Fi transmitter. This enables the transmission of data through VLC. The Li-Fi transmitter uses the LED bulb to transmit the data by modulating the light at very high speeds. The switching frequency of the LED must be high enough to avoid any flickering, which is crucial for ensuring the safety and comfort of human eyes. The rapid modulation allows the LED to transmit data without perceptible changes in light intensity, maintaining a consistent illumination while still

enabling data communication. Figure 5.illustrates the working model of the transmitter for the HMS utilizing VLC. This figure provides a detailed visual representation of how the system components interact, showcasing the integration of sensors, the PIC microcontroller, and the Li-Fi transmitter in the overall design of the transmitter section.

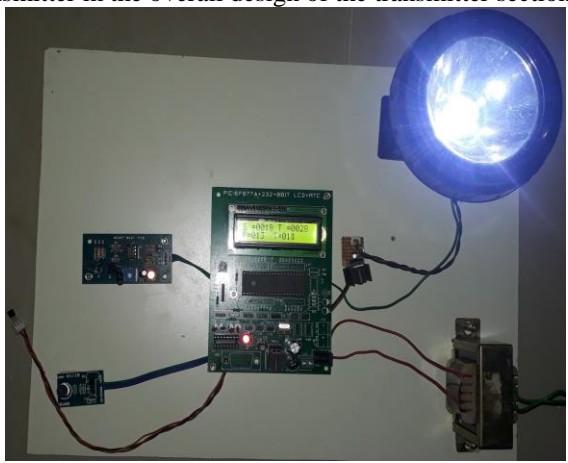

Figure 5. Working model of transmitter for HMS using VLC

*C. Receiver section:*

This section is also known as the monitoring section because the patient's results are monitored continuously through an LCD display. The monitoring section consists of several key components, including a Li-Fi receiver (photo detector), an Arduino UNO328P, an LCD, and a buzzer. A photodiode is used as the Li-Fi receiver. The photodiode functions as a light-to-electricity converter, detecting the light signals transmitted by the Li-Fi transmitter and converting them into corresponding electrical signals. However, the initial electrical signal generated by the photodiode tends to be weak and noisy. to address these issues, the signal undergoes processing through several stages. First, it passes through signal processing and amplification units to strengthen the signal and reduce noise. Following amplification, an envelope detector is used to demodulate the signal. The envelope detector extracts the original data signal from the modulated carrier wave. Next, a low-pass filter is employed to remove high-frequency noise from the signal, ensuring that the resulting signal is clean and suitable for further processing. Once the signal has been filtered, it is fed into a voltage comparator. The voltage comparator transforms the analog signal into a digital format, making it compatible with digital processing systems. The digital signal is then passed to the Arduino UNO328P for further processing. The Arduino microcontroller analyzes the incoming data and displays the relevant information on the LCD screen, providing real-time monitoring of the patient's health parameters. Additionally, the system can trigger a buzzer to alert healthcare personnel in case of critical readings or emergencies. Figure 6 illustrates the proposed block diagram for the receiver section of the HMS utilizing VLC. This diagram visually represents the flow of data from the Li-Fi receiver to the Arduino, highlighting the key components involved in signal processing and data presentation.

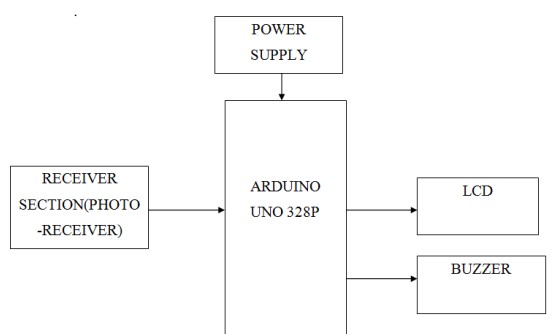

Figure 6 block diagram of receiver

For the system to function effectively, both the transmitter and receiver must be in a line of sight (LOS) position. This requirement ensures that the light signals can be transmitted and received without any obstructions. The information received by the receiver can be depicted in digital form, allowing for detailed analysis of the patient's health by displaying the data on an LCD screen. Additionally, the buzzer is connected to provide alertness to caregivers about the patient's health condition, with alerts being issued every 15 seconds for respiration values and every 30 seconds for heartbeat values.

Figure 7 shows the working model of the receiver section, detailing the flow and processing of data from the Li-Fi receiver to the LCD display and the buzzer.

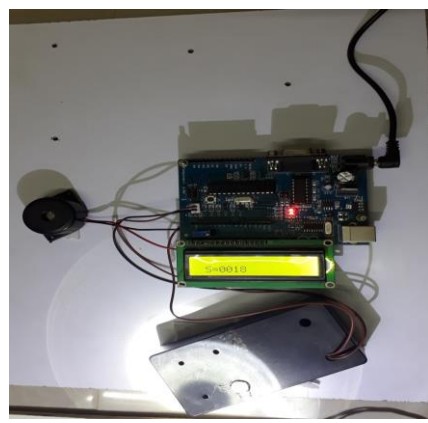

Figure 7 working model of receiver

Figure 8 shows the Transmitter and receiver section which contains LED as transmitting medium and photo detector as receiving medium. The sensors connected with the transmitting provide the details of health condition and those are transmitted through the light. This can be applied all medical fields which can be used without any interference.

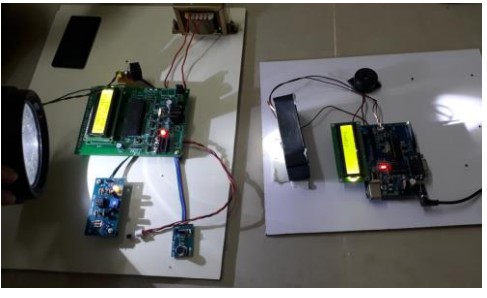

Figure 8 Hardware setup of HMS using VLC

TABLE 1. COMPARISON BETWEEN EXISTING RADIO FREQUENCY AND VISIBLE LIGHT COMMUNICATION

| Parameters | Visible light communication | Radio frequency |
|---|---|---|
| Distance coverage | Narrow | wide |
| Medium | Light illumination | Access point |
| Power | Based on LED | Low/medium |
| Security | Low | Enhanced |
| Electromagnetic interference | No | Yes |
| Multipath | Low (Line of sight) | High |

TABLE 2: COMPARISON OF OTHER WIRELESS COMMUNICATION TECHNOLOGIES WITH VLC.

| Wireless Technologies | Bandwidth | Line of sight | Power consumption | Coverage | Standards |
|---|---|---|---|---|---|
| VLC | 380-750 ns | Yes | LED consumes low power with illumination | limited | IEEE 802.15.7 |
| Infrared | Regulated from Radio frequency/ Limited | Yes | Low power | Short range | - |
| Bluetooth | Regulated from Radio frequency/ Limited | No | Low power | Short range | IEEE 802.15.1 |
| WIFI | Regulated from Radio frequency/ Limited | No | Average | limited | IEEE 802.15.11 |
| Zigbee | Regulated from Radio frequency/ Limited | No | Low power | Short range | IEEE 802.15.4 |

Table 1 and 2 list the existing technologies used for signal transmissions each with its own characteristics. All these technologies uses radio frequency signals, which can leads for RF congestion. The proposed VLC can be used as an alternative way for data transmission, serving both as communication medium and as source illumination.

Limitations:
- **Line of sight**: The transmitter and receiver must maintain a perfect line of sight to effectively transmit and receive data. Any obstruction can disrupt the communication.

- **Limited range:** The range of the light beams used in Li-Fi technology is relatively short, typically about 5 to 10 meters. This limits the distance over which data can be transmitted effectively.
- **Device compatibility:** Li-Fi technology only works on devices that are equipped with a Li-Fi receptor. This means that not all tablets, smartphones, and other devices can utilize Li-Fi without the necessary hardware.
- **Infrastructure Requirement:** Implementing Li-Fi technology requires the construction of a whole new infrastructure. This includes installing LED bulbs capable of transmitting data and ensuring all relevant devices have the necessary receptors.
- **Limited Penetration:** Light signals used in Li-Fi cannot penetrate through bricks or walls. As a result, Li-Fi can only be used within a single room, limiting its application in larger or multi-room environments.

## IV. CONCLUSION

Li-Fi is emerging as a more suitable network for next-generation healthcare services in hospitals. Patient monitoring can be done efficiently using Li-Fi technology, which offers several advantages over traditional communication methods. In this paper, we demonstrated the application of Visible Light Communication (VLC) in a Health Monitoring System using a prototype model. It is shown that a Li-Fi network can be successfully utilized as a high-speed, secure, and safe method for data communication, providing real-time monitoring of vital signs such as heartbeats, respiration, and temperature. Li-Fi technology significantly reduces radio interference in the human body, an important consideration in sensitive healthcare environments. The system measures the patient's data automatically and continuously, ensuring constant monitoring without manual intervention. In the future, this system can be expanded to monitor multiple patients simultaneously. Each LED bulb in the hospital can serve as a monitoring point for a patient, leveraging the widespread presence of lighting infrastructure. Using this technology in the medical field offers several benefits, including faster diagnosis and the ability to access the internet alongside devices that use radio waves. The proposed system is fully automated, which means it operates without the need for continuous human oversight, enhancing efficiency and reliability. If successfully implemented, this system could represent a significant milestone in the medical field, revolutionizing the way patient monitoring is conducted and improving overall healthcare delivery.

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
