# OpenReview forum: "Design and Implementation of Telemedicine System Using Light Fidelity and PIC16F877A Microcontroller"
_IEEE.org/ICIST/2024/Conference — IEEE ICIST 2024 Conference Submission_

### Official Review · Reviewer_e5e6 · 2024-08-26
**Design and Implementation of Telemedicine System Using Light Fidelity and PIC16F877A Microcontroller**

**Rating:** 5
**Confidence:** 2

**Review:**

This paper demonstrates the application of Visible Light Communication in a Health Monitoring System using a prototype model. The authors achieve valuable research results. However, this paper should be revised according to the following suggestions.
1. To put the contributions in context of the most recent research, please make sure the Related Work section contains relevant references that have appeared in the past two years in conferences and journals.
2. Please compare the results with previous methods. Experiments should include numerical comparisons with previously published methods or a convincing explanation why no previous methods are relevant.
3. The authors should standardize the format of figure titles. In addition, Figures 3 and 5 should be adjusted to the appropriate size.

---

### Official Review · Reviewer_XSjY · 2024-08-27
**This paper is a good paper, which can be accepted.**

**Rating:** 8
**Confidence:** 4

**Review:**

This paper investigated the problem of a novel design and implementation of a medical healthcare information system by using the newly developed wireless visible light communication (VLC) technology.
1) In abstract, about the proposed method, the statement is unclear. Authors need to rewrite abstract and to focus on the proposed method and to stress both the specific application and the generic aspects of the work.
2) The footer in the page 1:“XXX-X-XXXX-XXXX-X/XX/$XX.00 ©20XXIEEE”, should be
removed.
3) The size of pictures 3,5 and 8 needs to be adjusted.

---

### Official Review · Reviewer_xMev · 2024-08-30
**Accept**

**Rating:** 6
**Confidence:** 5

**Review:**

This paper proposes a novel design and implementation of a medical healthcare  information system by VLC, and wireless data transmission is realised through lighting fixtures, which solves the problem of limited radio frequency in the hospital and improves the safety of medical treatment. There are some suggestions:
1. The innovation of this paper can be expressed more explicitly.
2. The format of the paper could be improved.

---

### Decision · Program_Chairs · 2024-09-08

Accept (Oral)